# Dietary Catalase Supplementation Alleviates Deoxynivalenol-Induced Oxidative Stress and Gut Microbiota Dysbiosis in Broiler Chickens

**DOI:** 10.3390/toxins14120830

**Published:** 2022-11-28

**Authors:** Weiwei Wang, Jingqiang Zhu, Qingyun Cao, Changming Zhang, Zemin Dong, Dingyuan Feng, Hui Ye, Jianjun Zuo

**Affiliations:** Guangdong Provincial Key Laboratory of Animal Nutrition Control, College of Animal Science, South China Agricultural University, Guangzhou 510642, China

**Keywords:** antioxidant property, catalase, deoxynivalenol, gut microbiota, intestinal health

## Abstract

Catalase (CAT) can eliminate oxygen radicals, but it is unclear whether exogenous CAT can protect chickens against deoxynivalenol (DON)-induced oxidative stress. This study aimed to investigate the effects of supplemental CAT on antioxidant property and gut microbiota in DON-exposed broilers. A total of 144 one-day-old Lingnan yellow-feathered male broilers were randomly divided into three groups (six replicates/group): control, DON group, and DON + CAT (DONC) group. The control and DON group received a diet without and with DON contamination, respectively, while the DONC group received a DON-contaminated diet with 200 U/kg CAT added. Parameter analysis was performed on d 21. The results showed that DON-induced liver enlargement (*p* < 0.05) was blocked by CAT addition, which also normalized the increases (*p* < 0.05) in hepatic oxidative metabolites contents and caspase-9 expression. Additionally, CAT addition increased (*p* < 0.05) the jejunal CAT and GSH-Px activities coupled with T-AOC in DON-exposed broilers, as well as the normalized DON-induced reductions (*p* < 0.05) of jejunal villus height (VH) and its ratio for crypt depth. There was a difference (*p* < 0.05) in gut microbiota among groups. The DON group was enriched (*p* < 0.05) with some harmful bacteria (e.g., *Proteobacteria*, *Gammaproteobacteria*, *Enterobacteriales*, *Enterobacteriaceae*, and *Escherichia*/*Shigella*) that elicited negative correlations (*p* < 0.05) with jejunal CAT activity, and VH. DONC group was differentially enriched (*p* < 0.05) with certain beneficial bacteria (e.g., *Acidobacteriota*, *Anaerofustis*, and *Anaerotruncus*) that could benefit intestinal antioxidation and morphology. In conclusion, supplemental CAT alleviates DON-induced oxidative stress and intestinal damage in broilers, which can be associated with its ability to improve gut microbiota, aside from its direct oxygen radical-scavenging activity.

## 1. Introduction

Deoxynivalenol (DON), also known as vomitoxin, is a secondary metabolite of *Fusarium graminearum*. As one of the most severe mycotoxins prevalent in multifarious crops, especially grains [1,2], DON can elicit serious detriments to animal growth and health when diets are prepared with DON-contaminated grains [3,4]. Animals exposed to DON may exhibit acute or chronic poisoning, as manifested by the structural and functional injuries of multiple organs (e.g., liver, kidney, and intestine), accompanied with a series of clinical symptoms, such as vomiting, anorexia, diarrhea, and intestinal bleeding [5,6]. Despite the lesser susceptibility to DON exposure than other monogastric animals (e.g., pigs), poultry exposed to DON can still display metabolic abnormalities and health disorders [5,6].

One of the toxicological mechanisms of DON for animals is to cause mitochondrial dysfunction, leading to increased generation of reactive oxygen radicals, concurrent with decreased production of antioxidants in the cells [6,7], which can break the redox balance inside the body [8]. Once the accumulation of oxygen radicals exceeds their elimination, there can be oxidative stress responses within the body with a peroxidation and destruction of certain biomacromolecules, especially DNA and proteins [8], subsequently impairing animal growth and health [6,8]. Thereby, suppression of the inducive oxidative stress is a potential strategy to attenuate the toxic effects of DON in chickens.

Antioxidant enzymes play crucial roles in contributing to the scavenging actions of antioxidant system on various free radicals generated in the body [9]. Among the antioxidant enzymes, catalase (CAT) is a key member capable of catalyzing the decomposition of hydrogen peroxide (H_2_O_2_), which is a kind of reactive oxygen radicals and may trigger oxidative stress of the body when it exceeds the physiologic concentration [10]. More importantly, CAT can prevent reactions of H_2_O_2_ with oxygen under the action of iron chelates from producing more toxic hydroxyl radicals (·OH) and some other radicals [11]. Additionally, certain hydrogen donors, such as methanol and ethanol, may also be scavenged during the catalytic reaction of CAT, which can benefit the mitigation of oxidative stress [11,12]. Previous reports have shown that dietary CAT addition resulted in an enhancement of antioxidant property, with simultaneous improvements of growth and health performance in both broilers and pigs [13,14]. Moreover, CAT addition was reported to alleviate oxidative stress-induced intestinal and hepatic damages in pigs [15,16,17]. However, few studies are available concerning the potentially beneficial roles of CAT in alleviating the detriments of DON to chickens.

Gut microbiota elicit profound roles in mediating the impacts of dietary treatments on chicken growth and health [18]. There is evidence that dietary mycotoxin, including DON contamination-induced toxicity and oxidative damages in animals, were at least partially realized by the inducive disturbance of gut microbiota [19,20,21]. Growing studies implied that CAT addition had an ability to enhance antioxidant capacity, as well as improve intestinal and hepatic health, through associating with an optimization of gut microbial composition in both broilers [14] and pigs [16,22]. Nevertheless, the protective effects of CAT on the antioxidant property and gut microbiota in chickens exposed to DON remain unclear. Accordingly, the present study aimed to investigate the potential roles of supplemental CAT in alleviating DON-induced oxidative stress and gut microbiota dysbiosis in broiler chickens.

## 2. Results

### 2.1. Growth Performance and Organ Indexes

As shown in Table 1, there were no differences (*p* > 0.10) in the initial body weight (IBW) and final body weight (FBW), average daily gain (ADG), and average daily feed intake (ADFI) coupled with feed conversion ratio (FCR) among groups. However, the DON group showed a decreasing trend (*p* = 0.093) of average daily feed intake (ADFI), relative to the control or DONC groups. Regarding the organ indexes (Table 2), the indexes of spleen and bursa of Fabricius showed no differences (*p* > 0.10) among groups. The liver index and kidney index in the DONC group were lower (*p <* 0.05) and tended to be higher (*p =* 0.078), respectively, than those of the DON group, but similar to (*p* > 0.10) the control group.

### 2.2. Oxidative Status of the Intestine and Liver

As shown in Table 3, the contents of oxidative metabolites, including reactive oxide species (ROS), superoxide anion (O_2_^−^), hydroxyl radical (OH), 8-hydroxy-2′-deoxyguanosine (8-OHdG), and malondialdehyde (MDA) in the jejunum, did not differ (*p* > 0.05) among groups. However, the DON group showed increases (*p <* 0.05) in hepatic ROS, 8-OHdG, and MDA contents, relative to the control group. Supplemental CAT normalized hepatic ROS and 8-OHdG contents in DON-exposed birds to levels similar (*p* > 0.05) to the control group, and it abolished (*p <* 0.05) the DON-induced increase in hepatic MDA content. Regarding the antioxidant indicators, jejunal CAT and GSH-Px activities, along with hepatic T-AOC, were detected to be lower (*p* < 0.05) in the DON group versus control group. However, supplementing CAT to DON-exposed birds attenuated (*p <* 0.05) the decreased activities of jejunal CAT and GSH-Px and increased (*p <* 0.05) jejunal T-AOC.

### 2.3. Relative mRNA Expression of Antioxidation- and Apoptosis-Related Genes

As exhibited in Figure 1, the DON group displayed increases (*p* < 0.05) in the relative expression levels of nuclear factor erythroid-2 related factor (Nrf2) and Bcl-2 associated X protein (Bax) in the jejunum, together with heme oxygenase 1 (HO-1) and Caspase-9 in the liver, as compared with the control group. Supplementing CAT to DON-exposed birds did not change (*p* > 0.05) the relative expression of jejunal Nrf2 and Bax or hepatic HO-1. However, DON-exposed birds supplemented with CAT supported a similar (*p* > 0.05) expression of hepatic Caspase-9, relative to the control group.

### 2.4. Intestinal Morphological Structure

As presented in Figure 2A, jejunal villi in the control group were straight with a completed structure, while the DON group had a destruction of jejunal morphological structure, as evidenced by the breakage, shedding, and shortening of villi. However, the above phenomena were alleviated when birds were supplemented with CAT. Concretely, the broilers in the DON group had reduced (*p <* 0.05) villus height (VH) and villus height to crypt depth ratio (VCR), rather than crypt depth (CD) of the jejunum, compared with those in control group (Figure 2B–D). Nevertheless, the DON-induced reduction of jejunal VCR was reversed (*p <* 0.05) by CAT addition, which also rendered jejunal VH of DON-exposed birds, comparable (*p* > 0.05) to that in the control birds.

### 2.5. Gut Microbiota

#### 2.5.1. Diversity of Gut Microbiota

No difference (*p* > 0.05) was noted in the α-diversity indexes of gut microbiota among the groups (Appendix A). β-Diversity analysis manifested a difference (*p* < 0.05) in the similarity of gut microbiota among the groups (Figure 3A). This could be visualized by partial least squares discriminant analysis (PLS-DA) plot (Figure 3B), which revealed a distinct separation of microbiota among the groups.

#### 2.5.2. Gut Microbial Composition

As shown in Appendix A, the predominant phylum of broiler gut was *Firmicutes*, followed by *Proteobacteria*. Within *Firmicutes*, the majority belonged to the classes *Clostridia* and *Bacilli*, while the majority within *Proteobacteria* were *Gammaproteobacteria*. Orders level analysis showed that the gut of the control group was mainly occupied by *Oscillospirales* and *Lachnospirales*, while the DON and DONC groups were dominated by *Oscillospirales*, *Lachnospirales*, and *Enterobacteriales*. At the family level, the major members in control group were *Lachnospiraceae* and *Ruminococcaceae*, while those in the DON and DONC groups were *Lachnospiraceae*, *Ruminococcaceae*, and *Enterobacteriaceae*. The dominating genera in the control group were *Faecalibacterium* and the unclassified *Lachnospiraceae*, while those in DON group were *Escherichia*/*Shigella* and the unclassified *Lachnospiraceae*. In comparison, DONC group was dominated by *Escherichia*/*Shigella* and *Ruminococcus torques group*.

#### 2.5.3. Bacterial Richness among Groups

Bacterial richness (*p* < 0.05, linear discriminant analysis score (LDA) > 2.0) was identified by LDA combined effect size measurements (LEfSe) analysis. As illustrated in Figure 4, certain bacterial members, such as phylum Firmicutes, class Clostridia, orders Oscillospirales and Peptococcales, and families Ruminococcaceae and Peptococcaceae, together with genera Faecalibacterium, Flavonifractor, and Paludicola, were detected to be enriched in control group. Strikingly, the phylum Proteobacteria, class Gammaproteobacteria, order Enterobacteriales, family Enterobacteriaceae, and genus Escherichia/Shigella were enriched in the DON group. In comparison, the DONC group was differentially enriched with phylum Acidobacteriota, order Eubacteriales, and family Anaerofustaceae, along with genera Anaerofustis and Anaerotruncus.

#### 2.5.4. Correlations of Gut Microbiota with Other Intestinal Parameters

Spearman’s correlation analysis was used to identify associations between gut microbiota and other intestinal parameters among groups. As shown in Appendix A, there were no correlations between gut microbiota and intestinal gene expression. However, the abundance of phylum Firmicutes showed a positive correlation (*p* < 0.05) with jejunal CAT activity and VH (Figure 5), whereas a contrasting pattern was found for the phylum Proteobacteria and its affiliate members, including class Gammaproteobacteria, order Enterobacteriales, family Enterobacteriaceae, and genus Escherichia/Shigella. The abundance of phylum Acidobacteriota had a positive correlation (*p* < 0.05) with jejunal T-AOC. The class Clostridia, order Peptococcales, and genus Flavonifractor were positively correlated (*p* < 0.05) with jejunal CAT activity, and the order Oscillospirales was positively correlated (*p* < 0.05) with jejunal CAT activity and VCR. Additionally, the genus Faecalibacterium elicited a positive correlation (*p* < 0.05) with jejunal VH.

## 3. Discussion

The presence of a control + CAT group is not necessary for this study, based on the following two reasons: (1) this study was aimed to evaluate the efficacy of CAT in alleviating DON-induced detriments to broilers; (2) the beneficial effects of CAT addition on the growth and antioxidant capacity of both broilers and pigs under normal status had been confirmed in either our previous experiments or other studies [13,14].

DON contamination of feedstuffs, especially grains, has been an increasingly common problem, as it can cause growth retardation in animals [1,5]. There is evidence that poultry are far less sensitive to DON, relative to other monogastric animals (e.g., pigs), based on a previous study in which poultry showed no clinical response to a DON level of less than 20 mg/kg DON in diets, whereas 1–2 mg/kg DON caused toxicity in pigs [5]. Similarly, Dersjant-Li et al. [23] reported that dietary DON level below 16 mg/kg elicited marginal impact on broiler growth. Azizi et al. [24] recorded little impairment except for a reduction of feed intake during the early stage of growth of broilers when fortifying DON in diet at 10 mg/kg. In this study, dietary DON fortification at a subclinical level (an estimate of 7 mg/kg) caused little compromise of broiler growth performance except for a decreasing trend of ADFI. Supplemental CAT failed to improve growth performance of broilers exposed to DON, which did not agree with the study of Tang et al. [14] who found increases in weight gain, feed intake and feed efficiency of non-challenged broilers in response to CAT addition. The discrepancy might be due to the difference in raising condition of broilers.

Organ index namely the relative organ weight is a common indicator of health status in chickens. The effect of DON contamination of diets on broiler organ index revealed in previous studies is quite conflicting. For example, broilers fed a diet containing 5 mg/kg DON was reported to have negligible change in organ index [25], whereas it was also manifested to increase spleen index [26]. When fortifying DON at 10 mg/kg in diet, broilers were found to show an increase in jejunum index and a reduction of the index of bursa of Fabricius in the study of Awad et al. [27] and Wu et al. [28], respectively, with no changes in the indexes of other organs. The highly variable outcomes in these studies might be associated with the exposure time of DON and broiler breeds. In this study, we recorded an increase in the liver index, with a decreasing trend of kidney index in the broilers exposed to DON at a subclinical dosage (an estimate of 7 mg/kg). The increase in liver index might be ascribed to the fact that the increased free radicals induced by DON exposure perturbed protein synthesis via attack on ribosome and subsequently triggered a compensatory swelling of liver (the main site of protein synthesis) [8,29], while the decreasing trend of kidney index was presumably due to the inductive atrophy of kidney [30]. At present, no published study was available regarding the effects of CAT on organ index in animals. Herein, we noted that supplemental CAT abolished the increase in liver index and tended to reverse the decreasing trend of the kidney index of DON-exposed broilers. It was possible that the elimination of free radicals upon CAT addition resulted in less injuries of liver and kidney [22,31], thereby attenuating liver swelling and kidney atrophy.

The resultant redox imbalance of broiler organs, such as the intestine and liver, resulting from DON exposure have been well-established [5,32]. Analogously, this study manifested that broilers exposed to DON exhibited a disturbance of redox status, as exhibited by the increased contents of hepatic oxidative metabolites (ROS, MDA, and 8-OHdG), as well as impaired antioxidant property (decreases in hepatic T-AOC with jejunal CAT and GSH-Px activities). It is well-established that ROS represent the major kinds of free radicals with the ability to attack the unsaturated fatty acids in cell membrane, which cause lipid peroxidation chain reaction and, finally, generate a large amount of MDA [33]. The 8-OHdG, a biomarker of DNA oxidative damage, is also produced by ROS attacking the carbon atom of guanine base in DNA [34]. CAT and GSH-Px can prevent from redox imbalance by specifically scavenging hydrogen peroxide and lipid hydroperoxides, respectively [10]. In this study, the increased production of oxidative metabolites, together with impaired antioxidant property, due to DON exposure, was deduced to cause oxidative stress and the biomacromolecular damage of broilers [5,7]. However, supplementing CAT to DON-exposed birds normalized the hepatic ROS and 8-OHdG contents, alleviated the increase in hepatic MDA content with decreases in jejunal CAT and GSH-Px activities, and increased jejunal T-AOC. These results highlighted that exogenous CAT was beneficial for attenuating DON-induced oxidative stress in broilers. Likewise, some previous studies reported that dietary CAT addition improved redox status by enhancing the activities of antioxidant enzymes (SOD, CAT, and GSH-Px) and lowering oxidative metabolite MDA in the liver and intestine of both broilers [14] and pigs [16].

In order to cope with the tissue damage induced by free radicals, the body has evolved a complex mechanism responding to oxidative stress, among which Keapl/Nrf2-ARE is the most important endogenous pathway regulating the redox status inside the body [34]. Keap1, a cytosol binding protein of Nrf2, can bind to Nrf2 to prevent it from entering the nucleus under normal status, thereby avoiding the increase of cell sensitivity to stressors [34]. However, the overproduction of free radicals may activate Nrf2, which is then released from Keap1, enters the nucleus, and interacts with the antioxidant response element (ARE), subsequently promoting the expression of a series of downstream antioxidants and eliminating excess free radicals [35]. HO-1 is a crucial antioxidant enzyme with a binding site of a promoter similar to that of ARE, thus being targeted by Nrf2 [34]. It was emphasized that DON exposure-induced oxidative stress, which could be mediated by the Nrf2/HO-1 pathway [35]. Similarly, the current study revealed an upregulation of jejunal Nrf2 and hepatic HO-1 expression in broilers, due to DON exposure, which could be a feedback response of the host defense mechanism to the inducive oxidative stress. Strikingly, CAT addition did not affect either Nrf2 or HO-1 expression in DON-exposed broilers, demonstrating that exogenous CAT probably moderated DON-induced oxidative stress by directly eliminating radical accumulation and subsequently reducing the consumption of antioxidant enzymes, instead of promoting the expression of antioxidant enzyme genes via the Nrf2/HO-1 pathway. DON-induced oxidative stress was known to closely involve in cell apoptosis [8], as supported by the findings that DON exposure increased the expression of several key pro-apoptosis genes (e.g., Bax and Caspase family proteins) and anti-apoptosis gene Bcl-2, as well as their ratio (Bax/Bcl-2) in the intestine and liver [36,37]. Herein, we observed that broilers exposed to DON displayed an increased expression of jejunal Bax and hepatic caspase-9 (a critical mediator of cell apoptosis), highlighting an initiation of cell apoptosis in the jejunum and liver of broilers, in response to DON contamination in diet. When supplemented with CAT, the increased expression of hepatic caspase-9 in DON-exposed broilers was normalized, demonstrating a potential of CAT addition to mitigate DON-induced hepatic cell apoptosis in broilers to some extent. This was similar to some previous studies in which CAT addition decreased the mRNA expression of Bax and its ratio to Bcl-2 expression in the liver, as well as the mRNA and protein levels of hepatic and intestinal caspase family proteins (caspase-3 and -9) in pigs confronted with an oxidative stress induced by lipopolysaccharide [16,22].

Oxidative stress in broilers resulting from dietary DON contamination has been documented to trigger a destruction of the intestinal structure [24,28]. As expected, the present study showed that DON exposure led to an impairment of the jejunal morphology, as evidenced by the breakage, shedding, and incompleteness of the villi, together with reductions in VH and VCR. These were likely connected with the detected redox imbalance of jejunum. However, supplementing CAT to DON-exposed broilers normalized jejunal VH and VCR and might, in turn, favor maintenance of intestinal absorption and barrier function. This benefit was speculated to be responsible by the observed protective effect of CAT addition on jejunal antioxidant enzyme activities in DON-exposed broilers, because it has evidenced that exogenous CAT ameliorated intestinal morphological structure, probably through associating with the simultaneous increase in intestinal antioxidant capacity in both broilers [14] and pigs [16].

Gut microbiota are well-known for exerting essential roles in regulating host growth and health. Increasing studies verified that DON exposure-induced oxidative damage was involved in the gut microbiota disturbance of animals [18,28]. On the other hand, CAT addition was reported to promote intestinal and hepatic health via an association with improvement of gut microbiota in animals [14,16,22]. Similarly, the PLS-DA plot in this study disclosed a distinct shift of gut microbiota of broilers following DON exposure; however, this shift was alleviated by CAT addition. Bacterial richness analysis supported changes in gut microbial composition among groups. Thereinto, the control group was enriched with certain beneficial bacteria, such as *Firmicutes*, *Clostridia*, *Oscillospirales*, *Peptococcales*, *Ruminococcaceae*, *Faecalibacterium*, and *Flavonifractor*. In general, *Firmicutes* and *Clostridia* are the predominating commensal bacteria in animal gut and encompass plentiful potentially beneficial bacteria, therefore benefiting intestinal health of host [38,39]. *Oscillospirales*, *Peptococcales*, *Faecalibacterium*, *Ruminococcaceae*, and *Flavonifractor* were characterized as producers of butyric acid [40,41,42], which has a strong ability to enhance the antioxidant properties of broilers [43]. In the current study, the abundances of *Firmicutes*, *Clostridia*, *Oscillospirales*, *Peptococcales*, *Faecalibacterium*, and *Flavonifractor* were positively correlated with intestinal CAT activity and/or VH, suggesting that the enrichments of these bacterial members in gut could be conducive to intestinal redox homeostasis of broilers in control group. Comparatively, several potentially pathogenic or harmful bacterial members, including *Proteobacteria*, *Gammaproteobacteria*, *Enterobacteriales*, *Enterobacteriaceae*, and *Escherichia*/*Shigella* were enriched in the DON group. *Proteobacteria* includes a mass of typical pathogens, such as *Salmonella*, *Shigella*, *Klebsiella*, and pathogenic *Escherichia coli* that can generate considerable toxins, thereby serving as a momemtous indicator of gut microbiota disturbance and health disorders of animals [44,45]. It has been documented that the expansions of *Proteobacteria*, *Gammaproteobacteria*, *Enterobacteriales*, *Enterobacteriaceae*, and *Escherichia*/*Shigella* in the gut could cause accumulation of lipopolysaccharide, rendering the occurrence of oxidative stress in animals [19,46,47,48,49]. Herein, we detected negative correlations of the abundances of *Proteobacteria*, *Gammaproteobacteria*, *Enterobacteriales*, *Enterobacteriaceae*, and *Escherichia*/*Shigella* with both intestinal CAT activity and VH, demonstrating that the enrichments of these bacteria in the DON group conduced the simultaneous intestinal oxidative damage of broilers. In contrast to the DON group, the DONC group was differentially enriched with several potentially beneficial bacteria, such as the phylum *Acidobacteriota*, along with the genera *Anaerofustis* and *Anaerotruncus*. It was indicated that *Acidobacteriota* might favor the suppression of intestinal oxidative stress, due to its connection with intestinal anti-inflammation and the antioxidation of animals [50,51]. Likewise, this study disclosed a positive correlation of the abundance of *Acidobacteriota* with intestinal T-AOC of broilers. *Anaerofustis* and *Anaerotruncus* in the gut were indicated to prompt fiber digestion and the production of short-chain fatty acids that might allow for improvements of the intestinal antioxidant properties and morphological structures of animals [28,52,53,54]. Accordingly, the intestinal enrichments of *Acidobacteriota*, *Anaerofustis*, and *Anaerotruncus* due to CAT addition were probably associated with the observed alleviation of the intestinal oxidative damage of broilers exposed to DON. This was similar to a previous study in which dietary CAT addition improved the intestinal antioxidant capacity and gut microbial composition in broilers [14].

## 4. Conclusions

Supplemental CAT had a capacity to attenuate oxidative stress and intestinal injury of broilers exposed to DON. It is possible that the improved gut microbial composition (reflected by the enrichments of several beneficial bacteria) following CAT addition contributed to the observed protection against DON-induced oxidative damage in broilers. Our findings provided a strategy for limiting the detriments of dietary DON contamination to poultry.

## 5. Materials and Methods

### 5.1. Animals and Experimental Design

The experimental animal protocols for the present study were approved by the Animal Care and Use Committee of the South China Agricultural University. A total of 144 one-day-old Lingnan yellow-feathered male broilers were randomly assigned to 3 groups, with 6 replicates per group and 8 birds per replicate. The initial body weight of birds was similar across replicates. Broilers fed a basal diet were considered the control group, and the other two groups received a DON-contaminated basal diet added with 0 (DON group) or 200 U/kg CAT (DONC group). The content of DON (1.3 mg/kg) in basal diet was estimated using an AgraQuant^®^ DON enzyme-linked immunosorbent assay (ELISA) kit (Romer Labs, Getzersdorf, Austria) with a Multiskan SkyHigh Microplate Reader (Thermo Fisher Scientific, Waltham, MA, USA). Dietary DON fortification was performed by supplementing DON-enriching rice to basal diet at the expense of corn. The final content of DON in diet was estimated at 7 mg/kg, which could be viewed as a subclinical dosage, as it was less than the clinical dosage (greater than 16 mg/kg) estimated in a previous study [23], but exceeded the maximum allowable dietary level (3 mg/kg), according to the Chinese Hygienical Standard for Feeds (GB13078-2017). CAT preparation (the theoretical value of enzyme activity was 200 U/g) was obtained from Vetland Bio-Technology Co., Ltd. (Shenyang, China). The actual activity of CAT in this supplement was determined to be 197 U/g. The supplemental CAT level was selected based on our preliminary experiment. Birds were housed in two-tier cages in an environmentally controlled room, in which the lighting program was 16 h per day, and room temperature was kept around 34 °C during the first three days and then gradually decreased to 24 °C on d 21. Birds had free access to the mash feed and water. The composition of basal diet is exhibited in Appendix A. At 21 day (d) of age, birds were randomly selected from each replicate (6 birds/group) for determination of growth performance and sample collection. After sacrifice of these birds, visceral organs, including the liver, spleen, kidney, and bursa of Fabricius and intestines were separated. The midpoint of jejunal section was removed and cleaved into two segments, one of which was fixed in 4% paraformaldehyde solution, while the other one was quick-froze by liquid nitrogen and reserved at −80 °C. Moreover, cecal digesta was collected for sequencing analysis of gut microbiota.

### 5.2. Fabrication of DON-Enriching Rices

DON-enriching rices were fabricated according to the following procedures: (1) the *Fusarium graminearum* strain PH-1, kindly provided by Prof. Chenglan Liu (College of Plant Protection, South China Agricultural University), was revived, the resulting hyphae were harvested and aerobically plated on potato dextrose agar medium at 25 °C for 7 d; (2) fresh rices were placed into conical flasks and soaked with pure water overnight, followed by autoclaved sterilization; (3) the plating medium containing *Fusarium graminearum* hyphae was split into smaller portions and evenly scattered on the sterile rice, followed by static culture at 25 °C for 3 d with a subsequent shake culture at 25 °C for 18 d under aerobic condition. The contaminated rice were then collected, dried at 50 °C, and smashed, and the contents of DON (117.12 mg/kg) and its acetylated derivatives 3-acetyl-DON (2.39 mg/kg) and 15-acetyl-DON (8.03 mg/kg) in the powder were determined using an AgraQuant^®^ DON ELISA kit (Romer Labs, Getzersdorf, Austria) and a liquid chromatography/mass spectrometry system (LCMS-8060, Shimadzu, Kyoto, Japan), respectively. The contents of some other mycotoxins (Appendix A) were also quantified using the corresponding ELISA kits (Romer Labs, Getzersdorf, Austria).

### 5.3. Determination of Growth Performance and Organ Indexes

Body weight and feed intake were recorded for each replicate on d 21 for calculating the average body weight (ABW) of broilers at 21 d of age, along with average daily gain (ADG), average daily feed intake (ADFI), and feed conversion ratio (FCR) during 1–21 d of age. The collected organs, including liver, spleen, kidney, and bursa of Fabricius were weighed for determination of organ indexes, as calculated by the ratio of organ weight (g) to body weight (kg).

### 5.4. Measurement of Oxidative Status

The liver and jejunum samples were separately homogenized with 1:9 (*w*/*v*) cold saline, followed by centrifugation in a high-speed refrigerated centrifuge (6380R, Eppendorf, Hamburg, Germany) at 6000 rpm for 10 min at 4 °C to obtain the supernatants. The levels of oxidative metabolites, including reactive oxygen species (ROS), superoxide anion (O_2_**^−^)**, hydroxyl radical (OH), 8-hydroxy-2′-deoxyguanosine (8-OHdG), and malondialdehyde (MDA), as well as the antioxidant indices, including the total antioxidant capacity (T-AOC), the levels of glutathione (GSH) coupled with the activities of CAT, total superoxide dismutase (T-SOD), and glutathione peroxidase (GSH-Px) in the supernatants were measured by micromethods using the corresponding kits provided by Meimian Bioengineering Institute (Nanjing, China) under the manufacturer’s instructions.

### 5.5. Measurement of Gene Expression

Total RNA from the jejunum and liver samples was isolated and purified using the FastPure Cell/Tissue Total RNA Isolation Kit (Vazyme Biotech. Co., Ltd., Nanjing, China) under the corresponding instructions. The concentration of isolated RNA was measured with a NanoDrop-2000 spectrophotometer (Thermo Fisher Scientific, Waltham, MA, USA). RNA purity was estimated by detecting the absorbance ratio at 260:280 nm. RNA integrity was verified via detection of the 18S and 28S bands in 1% agarose gel electrophoresis. Thereafter, RNA samples were reverse transcribed to cDNA samples by using the HiScript Ⅱ qRT SuperMix (Vazyme Biotech. Co., Ltd., Nanjing, China). Real-time PCR for examining gene expression was implemented in a CFX96Touch Real-Time PCR system (Bio-Rad Laboratories, Hercules, CA, USA) using the 2×Taq Master Mix (Vazyme Biotech. Co., Ltd., Nanjing, China). Reduced glyceraldehyde-phosphate dehydrogenase (GAPDH) acted as a reference gene. Information for the primers of GAPDH and target genes, including nuclear factor erythroid-2 related factor (Nrf2), heme oxygenase 1 (HO-1), B-cell lymphoma-2 (Bcl-2), Bcl-2 associated X protein (Bax), and caspase-9, are displayed in Table 4. The relative mRNA expressions of genes were calculated using the 2^−ΔΔCt^ method [55].

### 5.6. Analysis of Intestinal Morphological Structure

The fixed jejunal tissues were embedded in paraffin and stained with hematoxylin-eosin to obtain cross-sections. The intact and representative villi selected from each section were used for determining intestinal morphological structure with a light microscope. Villus height (VH) and crypt depth (CD) were defined as the height from villous tip to villus-crypt joint, respectively, based on which villus height to crypt depth ratio (VCR) was then calculated.

### 5.7. High-Throughput Sequencing of Gut Microbiota

Bacterial genomic DNA was extracted from cecal digesta using NucleoSpin^®^ DNA Stool kit (Macherey-Nagel company, Düren, Germany). The concentration and quality of extracted DNA were checked using Nanodrop 2000 (Thermo Fisher Scientific, Waltham, MA, USA) and gel electrophoresis. Bacterial 16S rDNA sequences spanning the variable regions (V3-V4) were amplified using primers 338F (5′-ACT CCT ACG GGA GGC AGC A-3′) and 806R (5′-GGA CTA CHV GGG TWT CTA AT-3′). The amplified products were paired-end sequenced on an Illumina Novaseq platform (Illumina, San Diego, CA, USA) at Biomarker BioTech. Inc. (Beijing, China). The effective reads were clustered into operational taxonomic units and classified at various taxonomic levels based on a 97% sequence similarity. Bacterial α-diversity was analyzed using the QIIME2 software, and bacterial β-diversity was assessed by the partial least squares discriminant analysis (PLS-DA). The differences in bacterial abundances among groups were detected using the linear discriminant analysis (LDA) combined effect size measurements (LEfSe) analysis. Spearman correlation analysis was used for detecting the correlations between bacterial composition and other parameters.

### 5.8. Statistical Analysis

Data are expressed as the mean ± standard deviation and analyzed by one-way ANOVA using the general linear model procedure of SPSS 20.0. Differences among groups were examined using Duncan’s multiple comparisons. Statistical significance was set at *p* < 0.05, and 0.05 ≤ *p* < 0.10 was thought as a tendency towards significance.

## Figures and Tables

**Figure 1 toxins-14-00830-f001:**
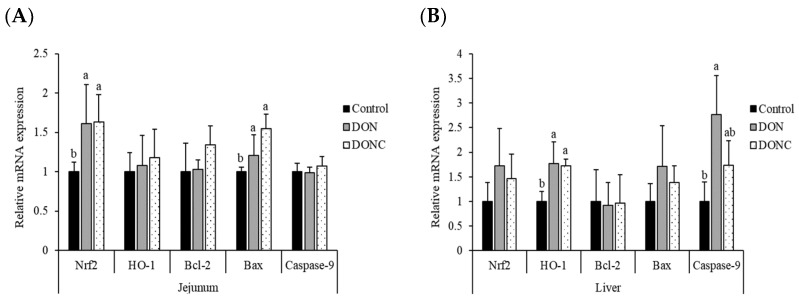
Effect of catalase on the relative mRNA expression of antioxidation-related genes (Nrf2 and HO-1) and apoptosis-related genes (Bcl-2, Bax and Caspase-9) in the jejunum (**A**) and liver (**B**) of broilers exposed to DON. Broilers in control and DON groups were fed a basal diet without and with DON contamination, respectively, while those in DONC group were fed a DON-contaminated basal diet supplemented with 200 U/kg catalase. ^a,b^ Values with unlike superscript letters differ significantly (*p* < 0.05). Nrf2, nuclear factor erythroid-2 related factor; HO-1, heme oxygenase 1; Bcl-2, B-cell lymphoma-2; Bax, Bcl-2 associated X protein.

**Figure 2 toxins-14-00830-f002:**
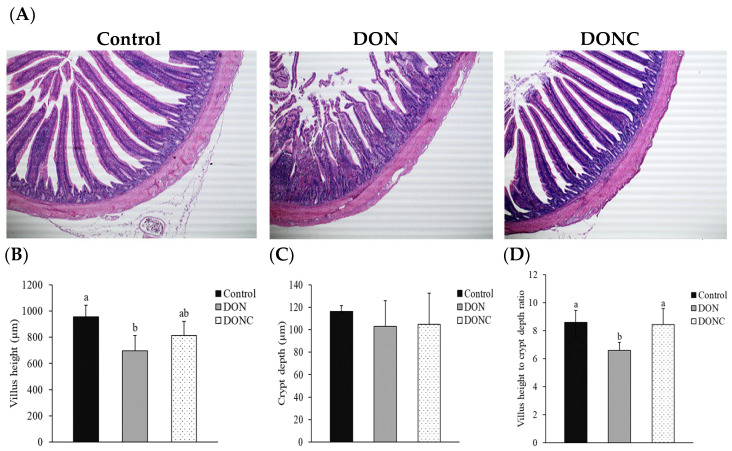
Effect of catalase on jejunal morphological structure of broilers exposed to DON. (**A**) Illustration (magnification 200×) of jejunal morphology of broilers from different groups. (**B**) Jejunal villus height of broilers from different groups. (**C**) Jejunal crypt depth of broilers from different groups. (**D**) Villus height to crypt depth ratio of the jejunum of broilers from different groups. ^a,b^ Values with unlike superscript letters differ significantly (*p* < 0.05). Broilers in control and DON groups were fed a basal diet without and with DON contamination, respectively, while those in DONC group were fed a DON-contaminated basal diet supplemented with 200 U/kg catalase.

**Figure 3 toxins-14-00830-f003:**
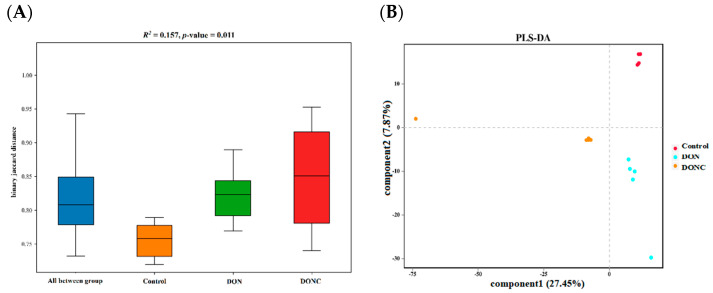
β-Diversity of broiler gut microbiota. (**A**) ANOSIM analysis (similarity analysis); (**B**) partial least squares discriminant analysis (PLS-DA). Broilers in control and DON groups were fed a basal diet without and with DON contamination, respectively, while those in DONC group were fed a DON-contaminated basal diet supplemented with 200 U/kg catalase.

**Figure 4 toxins-14-00830-f004:**
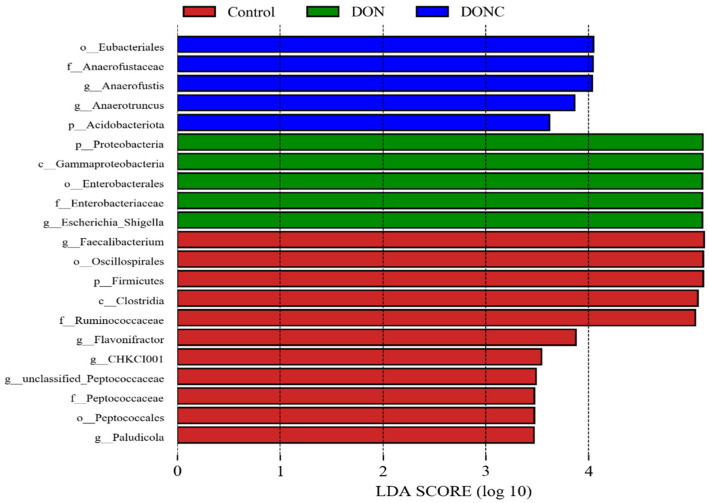
Linear discriminant analysis (LDA) combined effect size measurements (LEfSe) analysis of bacterial richness (*p* < 0.05, LDA > 3.0) in gut microbiota of broilers. Broilers in control and DON groups were fed a basal diet without and with DON contamination, respectively, while those in DONC group were fed a DON-contaminated basal diet supplemented with 200 U/kg catalase.

**Figure 5 toxins-14-00830-f005:**
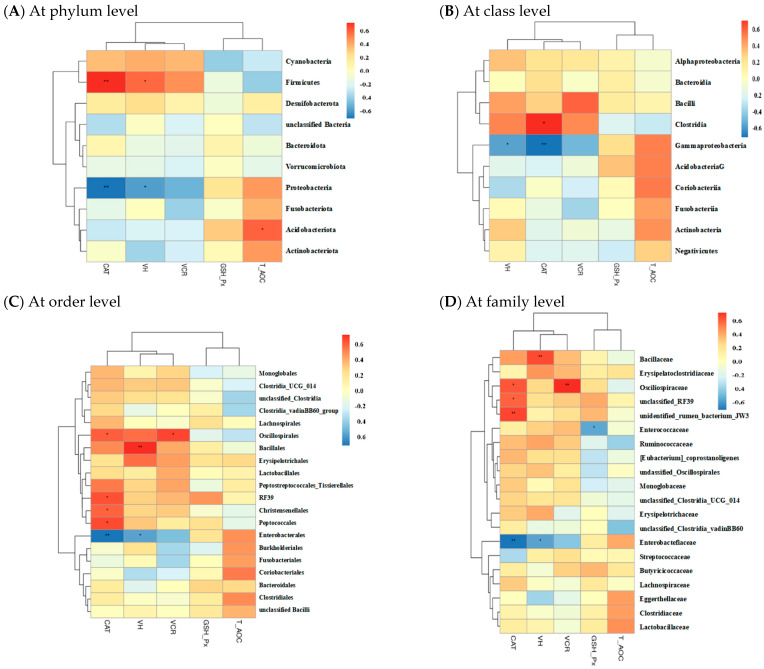
Correlation analysis of gut microbiota (**A**) at phyum level; (**B**) at class level; (**C**) at order level; (**D**) at family level; (**E**) at genus level) with intestinal antioxidant property and morphology in broilers. VH, villus height; CAT, catalase; VCR, villus height to crypt depth ratio; GSH-Px, glutathione peroxidase; T-AOC, total antioxidant capacity. The red and blue panes represent positive and negative correlations, respectively. Color intensity means the Spearman’s r-value of correlations in each panel. The asterisks indicate significant correlations (* *p* < 0.05; ** *p* < 0.01; *** *p* < 0.001). Broilers in control and DON groups were fed a basal diet without and with DON contamination, respectively, while those in DONC group were fed a DON-contaminated basal diet supplemented with 200 U/kg catalase.

**Table 1 toxins-14-00830-t001:** Effect of catalase on growth performance ^1^ of broilers exposed to deoxynivalenol (DON).

	Treatments ^2^			*p*-Value
	Control	DON	DONC	
IBW (g)	33.60 ± 0.21	33.50 ± 0.13	33.51 ± 0.17	0.593
FBW (g)	319.47 ± 22.41	309.83 ± 15.06	313.39 ± 13.59	0.635
ADG (g)	15.97 ± 1.12	15.49 ± 0.75	15.67 ± 0.68	0.636
ADFI (g)	31.53 ± 1.56	29.54 ± 1.38	30.77 ± 1.49	0.093
FCR	1.98 ± 0.12	1.91 ± 0.15	1.96 ± 0.13	0.676

^1^ IBW, initial body weight; FBW, final body weight; ADG, average daily gain; ADFI, average daily feed intake; FCR, feed conversion ratio. ^2^ Broilers in control and DON group were fed a basal diet without and with DON contamination, respectively, while those in DONC group were fed a DON-contaminated basal diet supplemented with 200 U/kg catalase.

**Table 2 toxins-14-00830-t002:** Effect of catalase on organ indexes ^1^ of broilers exposed to DON.

	Treatments ^2^			*p*-Value
	Control	DON	DONC	
Spleen (g/kg)	1.30 ± 0.33	1.52 ± 0.25	1.18 ± 0.45	0.419
Liver (g/kg)	23.66 ± 1.77 ^b^	28.56 ± 1.81 ^a^	25.16 ± 2.58 ^b^	0.016
Bursa of Fabricius (g/kg)	3.28 ± 0.96	3.21 ± 0.31	3.16 ± 0.41	0.957
Kidney (g/kg)	9.56 ± 0.71	8.46 ± 0.77	9.19 ± 0.95	0.078

^a,b^ Values within a row with unlike superscript letters differ significantly (*p* < 0.05). ^1^ Organ indexes were calculated as the ratio of organ weight (g) to body weight (kg). ^2^ Broilers in control and DON group were fed a basal diet without and with DON contamination, respectively, while those in DONC group were fed a DON-contaminated basal diet supplemented with 200 U/kg catalase.

**Table 3 toxins-14-00830-t003:** Effect of catalase on antioxidant parameters ^1^ of broilers exposed to DON.

		Treatments ^2^			*p*-Value
		Control	DON	DONC	
Jejunum	ROS	62.12 ± 5.94	64.88 ± 3.94	64.83 ± 7.56	0.667
	O_2_^−^	111.62 ± 48.46	115.78 ± 78.19	101.99 ± 35.02	0.924
	OH	6.51 ± 2.67	6.96 ± 1.80	5.46 ± 1.21	0.466
	8-OHdG	27.49 ± 0.97	29.02 ± 2.59	28.09 ± 1.77	0.394
	MDA	7.49 ± 0.22	7.30 ± 0.40	7.34 ± 0.31	0.327
	CAT	545.59 ± 29.53 ^a^	464.41 ± 8.66 ^c^	497.50 ± 11.86 ^b^	<0.001
	GSH-Px	114.79 ± 2.21 ^a^	111.18 ± 2.29 ^b^	117.75 ± 3.63 ^a^	0.008
	T-AOC	3.68 ± 0.31 ^b^	3.67 ± 0.23 ^b^	4.62 ± 0.65 ^a^	0.006
Liver	ROS	27.56 ± 0.52 ^b^	29.75 ± 1.32 ^a^	28.52 ± 1.34 ^ab^	0.031
	O_2_^−^	421.58 ± 35.68	445.83 ± 53.19	443.60 ± 73.66	0.753
	OH	33.26 ± 2.77	34.10 ± 6.16	33.16 ± 5.50	0.949
	8-OHdG	27.56 ± 0.52 ^b^	29.75 ± 1.32 ^a^	28.52 ± 1.34 ^ab^	0.031
	MDA	7.24 ± 0.086 ^b^	7.76 ± 0.20 ^a^	7.00 ± 0.26 ^b^	0.012
	CAT	554.118 ± 19.88	565.15 ± 18.03	545.41 ± 30.67	0.764
	GSH-Px	118.92 ± 5.68	123.71 ± 5.04	121.78 ± 3.50	0.255
	T-AOC	5.74 ± 0.19 ^a^	5.17 ± 0.42 ^b^	5.27 ± 0.07 ^b^	0.017

^a,b,c^ Values within a row with unlike superscript letters differ significantly (*p* < 0.05). ^1^ ROS, reactive oxygen species (ng/mL); O_2_^−^, superoxide anion (nmol/mL); OH, hydroxyl radical (ng/mL); 8-OHdG, 8-hydroxy-2′-deoxyguanosine (ng/mL); MDA, malondialdehyde (nmol/mL); CAT, catalase (U/mL); GSH-Px, glutathione peroxidase (U/mL); T-AOC, total antioxidant capacity (U/mL). ^2^ Broilers in control and DON group were fed a basal diet without and with DON contamination, respectively, while those in DONC group were fed a DON-contaminated basal diet supplemented with 200 U/kg catalase.

**Table 4 toxins-14-00830-t004:** Sequences for real-time PCR primers.

Genes ^1^	Primer Sequence ^2^ (5′-3′)	Accession No.
*GAPDH*	F: GAGGGTAGTGAAGGCTGCTG	NM_204305.1
	R: CATCAAAGGTGGAGGAATGG	
*Nrf2*	F: GATGTCACCCTGCCCTTAG	NM_205117.1
	R: CTGCCACCATGTTATTCC	
*HO-1*	F: GCTGAAGAAAATCGCCCAA	NM_205344.1
	R: ATCTCAAGGGCATTCATTCGG	
*Bax*	F: CAACAGGAAGAACACGCTGA	XM_015290060.1
	R: TCAGTCTCGGCCCACTATCT	
*Bcl-2*	F: GACAACGGAGGATGGGATG	NM_205339.2
	R: CAGGCTCAGGATGGTCTTCA	
*Caspase-9*	F: TCGAGCTGGCTCTGACATAGACTG	XM_424580.5
	R: AGGATGACCACGAGGCAGCAG	

^1^*GAPDH*: reduced glyceraldehyde-phosphate dehydrogenase, *Nrf2*: nuclear factor erythroid 2-related factor 2, *HO-1*: heme oxygenase 1, *Bax*: B-cell lymphoma-2, *Bcl-2:* Bcl-2 associated X protein. ^2^ F, forward; R, reverse.

## Data Availability

The datasets used in this study are available from the corresponding author upon reasonable request.

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
