# Peer review of "Dietary Catalase Supplementation Alleviates Deoxynivalenol-Induced Oxidative Stress and Gut Microbiota Dysbiosis in Broiler Chickens"

_toxins, 2022, doi:10.3390/toxins14120830_

Round 1

Reviewer 1 Report

This work describes the the benefits in reducing the impacts of Deoxynivalenol via the co-application of catalase. This manuscript is useful to the larger research community as it gives a better understanding of the impacts of DON on broiler chickens while also pointing towards a method of mitigation. 

There are 2 main points however that I feel greatly take away from the potential impact of this work

1) Although we have a control group, a DON group and a DON-catalase group, lacking another control group with only catalase administered makes it difficult to fully understand these results. For example, the spleen and the Bursa of Fabricus seem to have been impacted by the addition of catalase in an opposite direction as DON alone compared to the control.  Where the differences in these two metrics in the DON-catalase group significantly differently than the control group?

Also, please indicate why a catalase only control group was not included. Are there any previous publications where only catalase was applied in the feed and are those results relevant to the interpretation of these findings?

2- MAJOR CONCERN; - Re-analyze the feed material for the presence of additional toxins, specifically, acetylated DON and zearalenone. State if the F. graminearum strain used is a 15-acetyl- or 3-acetyl chemotype.

From the description of  how the DON was incorporated into the feed, it seems like whole contaminated rice material was used; DON was not purified and added to the diet. It is very well known that F. graminearum makes other toxins, specifically zearalenone.  See:

Seo, J. A., Kim, J. C., Lee, D. H., & Lee, Y. W. (1996). Variation in 8-ketotrichothecenes and zearalenone production by Fusarium graminearum isolates from corn and barley in Korea. Mycopathologia134(1), 31-37.

Megalla, S. E., G. A. Bennett, J. J. Ellis, and O. L. Shotwell. "Production of Deoxynivalenol and Zearalenone by Isolates of Fusarium graminearum Schw." Journal of food protection 50, no. 10 (1987): 826-828. USDA

Also, depending on the strain, it makes mostly acetylated DON precursor. Grown on a dead rice material, it is likely that the acetylated DON precursor may even be more abundant than DON itself. Therefore, the claims made in this paper, which associate all the results solely to DON exposure are likely not correct. ELISA is not sufficient since it does not detect zearalenone, and may be incorrectly cross-reacting with the acetylated DON present.

I believe it is absolutely critical that the authors re-analyze the contaminated rice material and report the concentrations of zearalenone and actylated DON. (and if it is 3-acetyl, or 15-acetyl).  Without this data, it is impossible to have confidence that the concentration of DON listed is the true concentration that was administered. Again, ELISA alone did not measure zearalenone nor can we be sure it did not cross-react with acetylated DON.

Minor comments:

 What is the specific ELISA assay kit that was used? In fact, all of the assay kits listed throughout need to be better described since ‘commericial kit’ is not sufficient. Many companies have several products. Please clarify this throughout the document

Line 31 – Fusarium graminae, or Fusarium graminearum.

Line 34 – DON usually may exhibit à DON may exhibit

Line 60 ‘Few study” ->  “few studies’

Table 1, header is ‘Treatments2’, should it be ‘Treatments (2 superscript)?

Author Response

Reviewer #1: There are 2 main points however that I feel greatly take away from the potential impact of this work.

  1. Although we have a control group, a DON group and a DON-catalase group, lacking another control group with only catalase administered makes it difficult to fully understand these results. For example, the spleen and the Bursa of Fabricus seem to have been impacted by the addition of catalase in an opposite direction as DON alone compared to the control. Where the differences in these two metrics in the DON-catalase group significantly differently than the control group? Also, please indicate why a catalase only control group was not included. Are there any previous publications where only catalase was applied in the feed and are those results relevant to the interpretation of these findings?

Answer: Although a 2 × 2 factorial arrangement design following a presence of control + catalase group may be more rigorous, the experimental design (one-factor random design) in the present study   can also be suitable to evaluate the efficacy of catalase in alleviating DON-induced detriments to broilers (the aim of this study) according to several previous studies (Xu et al. Toxins. 2022,14,641;  Awad et al. Poultry Science. 2006,85,974-979;  Wu et al. Journal of Animal Science and Biotechnology. 2018,9,74). What’s more, the beneficial effects of CAT addition on the growth and antioxidant capacity of both broilers and pigs under normal status had been confirmed in either our previous studies (published in Chinese journals) or other studies (Tang et al. Front. Vet. Sci. 2022, 9, 802051;  Guo et al. Animals. 2022,12, 828). Thereby, the presence of control + catalase group is not necessary for this study.               Regarding the results of organ indexes, neither the spleen nor the Bursa of Fabricus were impacted by the addition of catalase, as the P value of ANOVA is far greater than 0.05. Instead, we detected an increase in liver index with a decreasing trend of kidney index in broilers exposed to DON. The increase in liver index might be ascribed to that the increased free radicals induced by DON exposure perturbed protein synthesis via attack on ribosome and subsequently triggered a compensatory swelling of liver (the main site of protein synthesis) (You et al. Arch. Toxicol. 2021,95,1899-1915;  Chen et al. Poult. Sci. 2019,98,3802-3810.), while the decreasing trend of kidney index was presumably due to the inductive atrophy of kidney (Reddy et al. Toxins. 2018,10,114.).

  1. MAJOR CONCERN; - Re-analyze the feed material for the presence of additional toxins, specifically, acetylated DON and zearalenone. State if the F. graminearum strain used is a 15-acetyl- or 3-acetyl chemotype. From the description of how the DON was incorporated into the feed, it seems like whole contaminated rice material was used; DON was not purified and added to the diet. It is very well known that F. graminearum makes other toxins, specifically zearalenone.                 Also, depending on the strain, it makes mostly acetylated DON precursor. Grown on a dead rice material, it is likely that the acetylated DON precursor may even be more abundant than DON itself. Therefore, the claims made in this paper, which associate all the results solely to DON exposure are likely not correct. ELISA is not sufficient since it does not detect zearalenone, and may be incorrectly cross-reacting with the acetylated DON present.

Answer: Thanks for this comment. The F. graminearum strain used in this study was found to be only able to produce DON rather than zearalenone in the preliminary experiment. Regarding the potentially existed acetylated DON precursor, due to the absence of commercially available detection kits, we found it is not easy to be detected, which may require an analysis by liquid chromatography/mass spectrometer with corresponding expensive standard substances. This will certainly lead to an abvious increase in cost. Consistent with our study, previous studies regarding DON-exposed broiler models only detected the DON content in the feed material (rice and corn) and formula feed after contamination with F. graminearum (Wu et al. Journal of Animal Science and Biotechnology. 2018,9:74;  Yang et al. Journal of Animal Science. 2017.95:837-846. Qiu et al. Journal of Animal Science and Biotechnology. 2021,12:71). It can be thought that the acetylated DON may elicit an impact similar to that of DON on animals, despite that there may be differences in the impacts on broilers between acetylated DON and DON, and this may worth a further research.

Minor comments:

What is the specific ELISA assay kit that was used? In fact, all of the assay kits listed throughout need to be better described since ‘commericial kit’ is not sufficient. Many companies have several products. Please clarify this throughout the document. 

Answer: It has been modified throughout the text.

 Line 31 – Fusarium graminae, or Fusarium graminearum.

Answer: It has been corrected.

Line 34 – DON usually may exhibit à DON may exhibit

Answer: It has been modified.

Line 60 ‘Few study” ->  “few studies’

Answer: It has been modified.

Table 1, header is ‘Treatments2’, should it be ‘Treatments (2 superscript)?

Answer: It has been corrected.

Reviewer 2 Report

Dietary Catalase Supplementation Alleviates Deoxynivalenol induced Oxidative Stress and Gut Microbiota Dysbiosis in Broiler Chickens.

The Manuscript is not so well presented and lack proper proof reading. 

I also have some major questions regarding designing of experiments.

Statistical analysis not presented correctly in the text.

Authors should give more attention to explaining the originality of the work and why is it done rather than just reporting the results.

Few specific comments attached.

Author Response

Reviewer #2:

  1. The Manuscript is not so well presented and lack proper proof reading.

Answer: Thans for this comment. The manuscript have been re-proofread and modified.

  1. I also have some major questions regarding designing of experiments.

Answer:  It has been clarified in the following page.

  1. Statistical analysis not presented correctly in the text.

Answer:  It has been interpreted in the following page. There might be a misunderstanding and I have rephrased the corresponding contents.

  1. Authors should give more attention to explaining the originality of the work and why is it done rather than just reporting the results.

Answer:  It has been improved in the Discussion and Conclusion.

Specific comments

Line 13, 16 – Full form?

Answer: It has been modified.

Line 44-46 –Rephrase the sentence.

Answer: It has been done.

Line 82: ‘Why both..? Also no difference observed for IBW. So do mention it.

Answer: It has been modified.

Table 1: How was this p-value calculated. and why is it different than mentioned in the main text. Please check carefully as the tables are inconsistence with the text writing.

Answer: It has been checked. The p-value was made according to the ANOVA result. Differences among groups were examined using Duncan's multiple comparisons. Statistical significance was set at p < 0.05 and 0.05 ≤ p < 0.10 was thought as a tendency towards significance according previous studies (Wang et al. Animal Feed Science and Technology. 2020,264:114476;  Wang et al. Frontiers in Microbiology. 2020,11:600). We did not observe inconsistence in the p-value between Table 1 and the main text. There might be a misunderstanding and I have rephrased the main text.

Line 98: Why not to write DON, if once abbreviated.

Answer: It has been modified. The same as follows.

Line 99: Why and how this concentration was set?

Answer: The dose-response results in our preliminary experiment showed that addition of  200 U/kg CAT was suitable to produce benefits to broilers, besides, a previous study (Tang et al. Front. Vet. Sci. 2022,9:802051) also revealed obviously beneficial effects of supplemental 150~200 U/kg CAT on broiler antioxidant capacity and gut health. Thereby, we select this additive dose (200 U/kg) of CAT in this study.

Table 2: Again use abbreviation.

Answer: It has been modified. The same as follows.

Table 2: Again check for inconsistence with your own statement. Infact check it throughout the manuscript.

Answer: It has been checked. We did not observe inconsistence in the p-value between Table 2 and the main text. There might be a misunderstanding and I have rephrased the main text.

Figure 2: Provide scale on the picture.

Answer: It has been indicated.

Line 405-407: This sentence was repeated almost 6-7 times in the whole MS.

Answer: We have rephrased it.

Line 411-412: How is subclinical level determined?

Answer: According to previous studies (Dersjant-Li, et al. Nutrition Research Review. 2003, 16:223-239; Awad et al. Poultry Science. 2006, 85:974-979.), DON-induced clinical symptoms (e.g. growth depression or anorexia) of broilers can occur only when its contamination in diets at a dosage greater than 10 mg/kg. When the dosage of dietary DON is below 10 mg/kg but exceeds the Hygienical Standard for Feeds (3 mg/kg), broilers usually exihibit sub-clinical symptoms such as oxidative stress instead of clinical symptoms.  

Line 422: Why was it decided to feed them with fabricated rice and rather provide the toxin via water intake?

Answer: Firstly, this practice can make us to obtain the adequate amounts of DON, because the DON-producing fungi~Fusarium graminearum can quickly grow on cereal grains especially rices. Secondly, the detriments to chickens induced by mycotoxins including DON exposure is commonly through the feeds rather than drinking in production.

Line 467-469: Briefly provide the reason for selecting these specific genes? Any house keeping gene taken? How the gene expression was normalised?

Answer: 1) Keapl/Nrf2-ARE is well-known as the most important pathway regulating the redox status and protecting against DON-induced oxidative stress of animals. Activation of Nrf2 can promote the expression of a series of downstream antioxidants in which HO-1 is a crucial antioxidant enzyme to regulate the redox status. It has been suggested that DON-induced oxidative stress can be mediated by Nrf2/HO-1 pathway. DON-induced oxidative stress was previously reported to induce cell apoptosis of animal tissues through increasing the expression of several key pro-apoptosis genes such as Bax and Caspase-9 and decreasing the expression of key anti-apoptosis gene Bcl-2. The previous section of this study revealed that CAT could eliminate certain free radicals and/or increase the activities of several antioxidant enzymes in the liver and intestine, which could improve the redox status and might subsequently regulate the Nrf2/HO-1 pathway and cell apoptosis. Thereby, these specific genes were selected to be detected.

2) The reduced glyceraldehyde-phosphate dehydrogenase (GAPDH) was used as the house-keeping gene (namely reference gene) and its stability had been confirmed by measuring its expression abundance in each sample.

3) The method regarding the relative expression of target genes normalised to the reference gene have been well-described in many previous studies such as the study of Livak and Schmittgen (Methods. 2001,25:402-408). Briefly, the expression abundance (Ct value) of each target gene in each sample was firstly normalized to the corresponding abundance (Ct value) of house-keeping gene, which could obtain the ΔCt value (Ct target gene X - Ct reference gene) of each sample in each group, the average of ΔCt value in control group was then calculated as the normalizer. Afterwards, the ΔΔCt value of each sample in each group (ΔCt value of each sample in each group - the average of ΔCt value of control group) was calculated, followed by calculation of the value of 2-ΔΔCt (the relative mRNA expression) of each gene of each group.

Reviewer 3 Report

The Authors presented an article entitled “Dietary Catalase Supplementation Alleviates Deoxynivalenol- 2 induced Oxidative Stress and Gut Microbiota Dysbiosis in 3 Broiler Chickens.

Although the paper is interesting, I have some suggestions to improve the quality of paper:

General comments

1.      In Materials and methods section line 399 "….commercial enzyme-linked immunosorbent (ELISA) assay kit…."should be completed with the model of the Elisa equipment

2.      Line 445 " ….centrifugation at 6,000 rpm for 10 min at 4℃…."should be completed with the model of the centrifuge used.

3.      Figure 1- 3 – superabundant statement, the explanations should be written in text not in figure description.

Author Response

  1. In Materials and methods section line 399 "….commercial enzyme-linked immunosorbent (ELISA) assay kit…."should be completed with the model of the Elisa equipment

        Answer: It has been added.

  1. Line 445 " ….centrifugation at 6,000 rpm for 10 min at 4℃…."should be completed with the model of the centrifuge used.

      Answer: It has been added.

  1. Figure 1- 3 – superabundant statement, the explanations should be written in text not in figure description.

 Answer: Thanks for this comment. However, in general, the contents of      Figures must maintain its relative independence, the necessary informations in Figure capations can allow the readers to better understand the results. Similar practice could be found in some previous studies (Meerpoel el al. Toxins. 2020,12:719;  Silva-Cardoso et al. Toxins 2016, 8:316.).

Round 2

Reviewer 1 Report

Thank you for addressing most of my concerns.

I also agree in your explanation regarding why a 'catalase only' treatment was not required. Please add 1-2 sentences in the discussion explaining this, using the same reasons as listed in the response to my comments.

I still have a major issue with statements regarding the concentration of DON, and the absence zearalenone.

The authors used F.graminearum strain K082465.1 and stated that it did not produce zearalenone in an initial experiment. I could not find any additional information about the mycotoxin production of this strain in the literature.

Can the authors include the initial experiment they used to determine no zearalenone was present. This and any relevant figures/tables should be placed in the supporting information. 

Finally, I believe that the feed material should be analyzed for acetylated precursors of DON.  Since this paper will generate information about potential levels that impact broiler chickens, it is important to know the true concentration of DON. If the ELISA kits are not measuring acetylated DON, but the acetylated precursors are impacting the chickens, it creates a disconnect of how this work can be interpreted, especially from a NOAEL perspective.

If the authors cannot do this test of the feed material, consider rephrasing the entire manuscript to indicate that other toxins may be present, and consider the concentrations of DON used in this study to be 'estimates' and not confident concentrations.

Author Response

Reviewer #1:

Comment 1: I also agree in your explanation regarding why a 'catalase only' treatment was not required. Please add 1-2 sentences in the discussion explaining this, using the same reasons as listed in the response to my comments.

Answer: It has been done.

Comment 2:

The authors used F.graminearum strain K082465.1 and stated that it did not produce zearalenone in an initial experiment. I could not find any additional information about the mycotoxin production of this strain in the literature.

Can the authors include the initial experiment they used to determine no zearalenone was present. This and any relevant figures/tables should be placed in the supporting information.

Finally, I believe that the feed material should be analyzed for acetylated precursors of DON.  Since this paper will generate information about potential levels that impact broiler chickens, it is important to know the true concentration of DON. If the ELISA kits are not measuring acetylated DON, but the acetylated precursors are impacting the chickens, it creates a disconnect of how this work can be interpreted, especially from a NOAEL perspective.

If the authors cannot do this test of the feed material, consider rephrasing the entire manuscript to indicate that other toxins may be present, and consider the concentrations of DON used in this study to be 'estimates' and not confident concentrations.

Answer: The finding that F. graminearum strain used in this study was only able to produce DON rather than zearalenone was obtained in a preliminary test. The data regarding the contents of some other mycotoxins including zearalenone in the rice powder were supplemented and exhibited in Table S2.

Regarding the potentially existing acetylated DON precursors, we have rephrased the entire manuscript to indicate that other mycotoxins may be present, and the concentrations of DON in this study were 'estimates'.

Reviewer 2 Report

The manuscript has improved substantially after the careful revision by the authors.

I still have one concern regarding the toxin estimation from Fungi. A simple LC-MS analysis would be rather more optimal to estimate the quantity of the DON rather than just Elisa test.

What about affect of other possible secondary metabolite effects on the entire experiment which should not be neglected as the authors are not using purified toxins and the synergistic effect (of crude extract) should not be omitted from such kinds of experiments.

Author Response

Reviewer #2:

Comment 1: I still have one concern regarding the toxin estimation from Fungi. A simple LC-MS analysis would be rather more optimal to estimate the quantity of the DON rather than just Elisa test.

Answer: Of course, LC/MS is the the most scientific method determing the DON content. However, due to the high cost and inconvenience, it may not be the suitable method to quantify DON in practice. Instead, ELISA method have been confirmed as an alternative to LC/MS to determine the content of DON in crops and foods in many published studies (Sugita-Konishi et al. Evaluation of three commercial ELISA kits for rapid screening of deoxynivalenol in unpolished wheat. Food Hygiene and Safety Science. 2004, 45(3):156-160.   Li et al. High-sensitive chemiluminescent elisa method investigation for the determination of deoxynivalenol in rice. Food Analytical Methods. 2015, 8(3):656-660.    Pleadin et al. correlation of deoxynivalenol and fumonisin concentration determined in maize by ELISA methods. Journal of Immunoassay & Immunochemistry. 2012, 33(4):414-421.).

Comment 2: What about affect of other possible secondary metabolite effects on the entire experiment which should not be neglected as the authors are not using purified toxins and the synergistic effect (of crude extract) should not be omitted from such kinds of experiments.

Answer: Thanks for this comment. The toxic metabolites of F. graminearum are famous as a category of mycotoxins. Previous studies regarding DON-exposed broiler models only detected the content of mycotoxin DON in the feeds and diets after contamination with F. graminearum (Wu et al. Journal of Animal Science and Biotechnology. 2018,9:74;  Yang et al. Journal of Animal Science. 2017.95:837-846. Qiu et al. Journal of Animal Science and Biotechnology. 2021,12:71). In this study, the DON and some other mycotoxins potentially produced by F. graminearum in the feed were determined, the results revealed a very low content of mycotoxins (Table S2) other than DON in F. graminearum-inoculated rices. Although there might be some DON precursors (such as acetylated DON with a high cost of determination) are usually very difficult to be  detected in the diet, their effects on animal growth and health could be similar to those of DON itself due to the similar mechanisms of their toxicity (Pestka. Animal Feed Science and Technology. 2007,137(3-4):283-298;  Yang et al. Toxicology in Vitro. 2020,66:104838). Of course, the specific future studiers may be deserved.
